# RobotSmith: Generative Robotic Tool Design for Acquisition of Complex Manipulation Skills

**Chunru Lin[1]***, **Haotian Yuan[1]***, **Yian Wang[1]***, **Xiaowen Qiu[1]**, **Tsun-Hsuan Wang[2]**,
**Minghao Guo[2]**, **Bohan Wang[3]**, **Yashraj Narang[4]**, **Dieter Fox[4]**, **Chuang Gan[1,5]**

[1]University of Massachusetts Amherst     [2]Massachusetts Institute of Technology
[3]National University of Singapore     [4]NVIDIA     [5]MIT-IBM Watson AI Lab

## Abstract

Endowing robots with tool design abilities is critical for enabling them to solve complex manipulation tasks that would otherwise be intractable. While recent generative frameworks can automatically synthesize task settings, such as 3D scenes and reward functions, they have not yet addressed the challenge of tool-use scenarios. Simply retrieving human-designed tools might not be ideal since many tools (e.g., a rolling pin) are difficult for robotic manipulators to handle. Furthermore, existing tool design approaches either rely on predefined templates with limited parameter tuning or apply generic 3D generation methods that are not optimized for tool creation. To address these limitations, we propose *RobotSmith*, an automated pipeline that leverages the implicit physical knowledge embedded in vision-language models (VLMs) alongside the more accurate physics provided by physics simulations to design and use tools for robotic manipulation. Our system (1) iteratively proposes tool designs using collaborative VLM agents, (2) generates low-level robot trajectories for tool use, and (3) jointly optimizes tool geometry and usage for task performance. We evaluate our approach across a wide range of manipulation tasks involving rigid, deformable, and fluid objects. Experiments show that our method consistently outperforms strong baselines in terms of both task success rate and overall performance. Notably, our approach achieves a 50.0% average success rate, significantly surpassing other baselines such as 3D generation (21.4%) and tool retrieval (11.1%). Finally, we deploy our system in real-world settings, demonstrating that the generated tools and their usage plans transfer effectively to physical execution, validating the practicality and generalization capabilities of our approach.[2]

## 1 Introduction

Tool use is a fundamental capability of intelligent agents [16, 49, 47], enabling them to extend their physical affordances to accomplish tasks that would otherwise be unattainable. In both biological and artificial systems, the ability to use tools is critical for performing complex interactions with the environment, particularly under morphological or task-specific constraints [60, 35, 13, 18]. For robots operating in unstructured settings, developing robust tool-use behaviors is essential for achieving general-purpose manipulation capabilities.

Recently, a number of generative frameworks for robotic manipulation have emerged [54, 51, 9, 53, 12, 20, 23, 32], demonstrating strong potential to scale up data collection and task generalization. However, these frameworks have overlooked tool-use scenarios, as they typically rely on retrieving

---

*denotes equal contribution

[2]Project page: `https://umass-embodied-agi.github.io/RobotSmith/`

39th Conference on Neural Information Processing Systems (NeurIPS 2025).

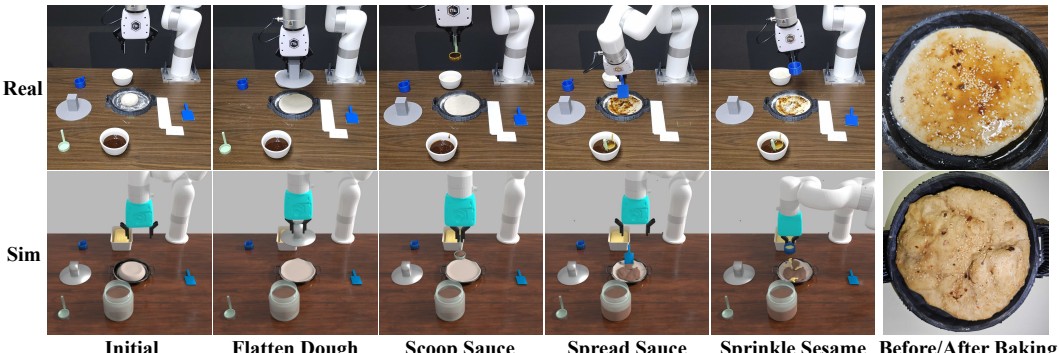

| Real | Sim |
| --- | --- |
| Initial | Flatten Dough | Scoop Sauce | Spread Sauce | Sprinkle Sesame | Before/After Baking |

Figure 1: Our pipeline enables fully autonomous long-horizon manipulation by generating and using tools for each subtask without human involvement. The robot completes the entire pancake-making process from shaping dough to spreading sauce and adding sesame. The bottom row shows simulated execution with generated tools, while the top row depicts the corresponding real-world stages of pancake preparation.

objects from datasets curated for human use rather than robotic manipulation. For example, a rolling pin might be selected for a dough-flattening task, but such a tool is difficult for a robotic arm to manipulate. A more suitable alternative for robots might be a custom-designed presser composed of a flat board and a handle as shown in Figure 1. Hence, shifting from using a predefined, often limited toolset to designing tools tailored to specific contexts, including different scenarios and robots, may unlock new possibilities for general-purpose manipulation.

Existing tool-design approaches fall short in addressing this gap. One line of work [35, 13, 18, 26] assumes predefined tool templates tailored for specific tasks and only optimizes the parameters within those templates. While efficient, such methods struggle to generalize beyond a narrow set of tool types, such as rigid-link structures. Another line of work explores tool generation by optimizing geometry in the latent space of 3D generative models [57, 19, 52]. However, these methods depend heavily on the quality and task alignment of the learned latent space, which is often not specifically designed for tool representation—limiting the scope and effectiveness of the generated tools.

To bridge this gap, there is a clear need for an automatic tool-design framework capable of generating effective, task-specific tools from scratch. Large pretrained models, such as vision-language models (VLMs), have demonstrated strong capabilities in spatial reasoning [63, 8, 10], affordance perception [60, 27], and commonsense physical inference [11, 55]—all of which are essential forms of prior knowledge for tool design. For example, spatial reasoning enables the model to determine how a tool should be shaped (e.g., connectivity and size) to interact effectively with objects in the environment. Affordance perception allows it to infer what geometric features are needed to enable specific functions such as scooping, pressing, or lifting. Commonsense physical inference further helps anticipate how the tool's geometry and structure will affect its physical interactions, ensuring it can successfully fulfill the task.

To this end, we propose **RobotSmith**, an automated framework that integrates the rich prior knowledge in large foundation models with a joint optimization process in physics simulation to enable both tool design and tool use for robotic manipulation. Given a specified manipulation task, RobotSmith generates a customized tool geometry, determines its placement within the scene, and synthesizes a corresponding manipulation trajectory for the robot. The framework is built around three core components:

1. *Critic Tool Designer*: Two collaborating VLM agents engage in iterative proposal and refinement to generate candidate tool geometries grounded in visual context and described in `json` format.

2. *Tool Use Planner*: Based on the designed tool and scene configuration, the system generates a manipulation trajectory using a minimal set of APIs to abstract away low-level control.

3. *Joint Optimizer*: The tool geometry and trajectory parameters are jointly fine-tuned through feedback from physics-based simulation to maximize task performance.

We evaluate the proposed framework across a suite of simulated manipulation tasks requiring nontrivial tool use, such as reaching an object outside the workspace, pouring water into a narrow-mouth bottle, cutting dough, and so on. Our method demonstrates superior performance relative to several baselines, including those using fixed tool libraries or generating tools directly. To assess physical feasibility and transferability, we additionally validate the approach in real-world settings: selected tools are fabricated using 3D printing, and their corresponding trajectories are executed on a physical robotic platform. The real-world experiments confirm that the generated designs and plans can be successfully realized in practice.

In summary, our key contributions are as follows:

- We propose a unified framework that integrates large foundation models with physics-based simulation to enable fully automated tool design and tool-use planning for complex robotic manipulation tasks. Our full pipeline, including code and APIs, will be made publicly available to support further research.
- We propose a parameterized, modular tool representation with a set of APIs for tool geometry synthesis. This enables a dual-agent, VLM-based designer to iteratively generate and refine tool designs using visual feedback, supporting precise, editable tools that ground high-level concepts in physically realizable geometry.
- We validate our approach in both simulated and real-world settings, demonstrating that it outperforms baseline methods and enables tasks that are otherwise difficult to solve without tools in simulation, improving the success rate from 2.8% to 50%, and confirming its physical plausibility through real-world executions across diverse manipulation tasks.

## 2 Related Works

### 2.1 Generative Pipelines in Robotics

Many recent works have explored building generative pipelines to automatically collect robotic manipulation data, offering the potential to scale up synthetic data generation for training and evaluation [54, 51, 9, 53, 12, 20, 23, 38]. These pipelines typically involve several key components, including task generation [54, 51, 23, 33, 50], scene synthesis [53, 54, 32, 41, 56], and reward function design [61, 5, 59, 46, 48, 36, 34, 62]. However, most existing efforts focus on simple, rigid-body tasks such as pick-and-place or door opening/closing, with limited attention to more complex scenarios like manipulating deformable materials. Such tasks often require specialized tools (e.g., spoons, dough pressers) that are adapted for robotic use, which current pipelines cannot automatically generate. Our pipeline aims to fill in this gap by automatically generating suitable tools for each given tasks.

### 2.2 3D Tool Design

Designing task-specific tools is labor-intensive and typically requires significant domain expertise. Prior work in tool design has generally targeted specific tasks using predefined tool templates [14, 2, 35, 13, 18], repurposed existing objects in the scene [28, 40, 60], or employed generative models to synthesize tools from scratch [57, 19]. These approaches optimize tool design using various methods, including differentiable physics [2], stochastic optimization [14], reinforcement learning [35, 13, 28], and supervised learning for generative frameworks [57, 19]. However, none of these methods are fully scalable or automated across diverse task settings. They are typically constrained by predefined templates or rely heavily on domain-specific data, limiting their ability to generalize to broader tool-use scenarios. In contrast, our method is designed to handle more diverse and complex situations by leveraging the prior knowledge embedded in foundation models and the robustness of physics-based simulation[4, 30].

### 2.3 Robotic Tool Use

Learning to use tools is a crucial capability for robots, significantly expanding the range of tasks they can perform. Prior research [25, 39, 17, 29, 24, 45] has explored teaching robots to use tools to accomplish complex tasks, such as manipulating chopsticks [25]. More recently, RoboCook [47] enabled a robot to make dumplings using tools, simplifying the challenging problem of deformable-

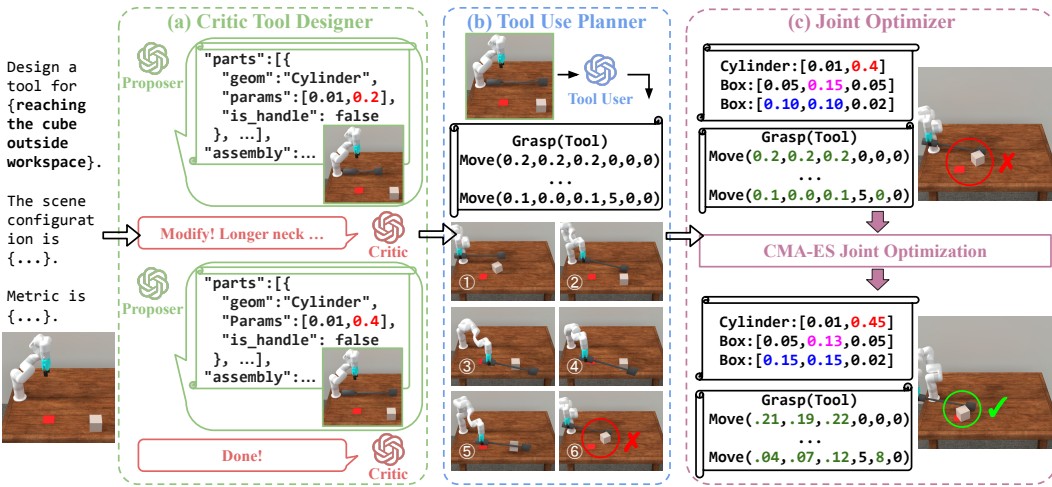

Figure 2: **Pipeline Overview.** The pipeline consists of three modules: (a) Critic Tool Designer, where two VLM agents iteratively propose and evaluate tool designs. The proposer takes in the user prompt, outputs a design, and renders an image, which is passed to the critic for feedback. This process continues until the critic is satisfied with the design; (b) Tool Use Planner that generates usage plans with a minimal set of high-level robot APIs; and (c) Joint Optimizer that fine-tunes tool geometry and trajectory based on task performance.

object manipulation and making it feasible for a robotic arm. These approaches typically rely on existing or human-designed tools and learn their use through affordance modeling [15, 43, 7, 44], dynamics prediction [47, 3, 58, 31], or guidance from large language models [60]. However, the approaches are often task-specific and assume the availability of suitable tools. Prior works [35, 13, 18] take a step further by jointly learning tool parameters and the policy for using them. Nonetheless, this method still requires a predefined tool template for each task. In contrast, our approach addresses tool design as part of the problem. Given a task that requires a tool, we jointly optimize both the tool's design and the corresponding robot trajectory, enabling the agent to reason about both aspects simultaneously.

## 3 Method

**Problem Setting.** Each manipulation task is defined as $(\mathcal{T}, \mathcal{S}_0, \mathcal{M})$, where $\mathcal{T}$ is a natural language task description (e.g., lift a bowl), $\mathcal{S}_0$ is the initial 3D scene configuration that specifies the objects and their spatial arrangement (e.g., `bowl=Mesh('bowl.obj', pos=(0,0.3,0), euler=(0,0,0))`), and $\mathcal{M}$ is a task-specific reward function (e.g., bowl height and no gripper contact). The system aims to produce $(\mathcal{G}, (p, e), \alpha)$, where $\mathcal{G}$ is a 3D mesh of the tool (detailed in Sec. 3.1), $(p, e) \in \mathbb{R}^3 \times \mathbb{R}^3$ is the initial placement of the tool, and $\alpha = \{a_1, a_2, ..., a_n\}$ is a robotic manipulation trajectory composed of high-level actions (described in Sec. 3.3). Task success is then evaluated using the metric $\mathcal{M}(\mathcal{S}_0 \cup (p, e), \mathcal{G}, \alpha)$, computed via physics-based simulation or physical execution.

We develop an automated framework that jointly addresses tool design and tool use for robotic manipulation tasks. The framework, which we refer to as **RobotSmith**, consists of three components: (1) Critic Tool Designer 3.2, (2) Tool Use Planner 3.3, and (3) Joint Optimizer 3.4. An overview is shown in Fig. 2 and Fig. 8. To support flexible and interpretable tool generation, we also introduce a modular and parameterized representation for tools in section 3.1.

### 3.1 Tool Representation

We design a modular and parameterized representation for tools that supports both procedural construction and parameterized refinement. Each tool consists of *a set of parts*, either basic geometric primitives (e.g., box, sphere, cylinder) or complex shapes generated by a text-to-3D model, along with size parameters.

To specify how individual parts are positioned and connected, we introduce *an assembly function*, a programmatic representation inspired by Constructive Solid Geometry (CSG)[22]. Tools are constructed as a tree of operations applied to parts, where each node corresponds to a geometric transformation, either unary (e.g., translation, rotation, scaling) or binary (e.g., union, subtraction, alignment). Although implicit, this CSG-like structure promotes valid tool construction by avoiding self-intersections and ensuring physical connectivity between components. By modularizing geometry and assembly logic, the representation supports flexible refinement: for example, directly shrinking a part may break connections with adjacent components, but the assembly function enforces alignment constraints, allowing the system to automatically reposition parts and maintain structural coherence. An example of our tool representation is in Appendix A.

Another key advantage of our representation is that all tool parts are parameterized. During the joint optimization stage, we can fine-tune parameters like size, position, or orientation to improve task performance without redesigning the tool from scratch. Moreover, by combining procedural primitives with generative 3D components, our framework can represent a broader range of tools than prior methods (Sec. 4.3), capturing both simple structures and semantically meaningful geometries.

## 3.2 Critic Tool Designer

The module consists of two collaborating vision-language model (VLM) agents: a *Proposer agent* and a *Critic agent* based on o3-mini[42]. These agents engage in an iterative design loop, as illustrated in Fig. 2 (a).

Specifically, the Proposer receives the task description and initial scene setup as input and proposes an initial tool design, which is then rendered into multi-view images passed to the Critic along with the design rationale. The Critic provides detailed feedback based on both task-specific requirements and user-defined structural principles; for example, ensuring that all parts are connected, that the tool is graspable, and that the tool is long enough to avoid undesired contact (e.g., with water). This feedback is returned to the Proposer, which refines the tool accordingly. This process repeats iteratively until the Critic returns a termination signal ("Done"), indicating that the design satisfies all key constraints. This approach offers an effective way to explicitly embed important design constraints and principles into the tool generation process.

## 3.3 Tool Use Planner

Once the Critic approves the tool design, it prompts a Tool User agent to generate an initial robot trajectory for using the tool, as shown in Fig. 2 (b). This trajectory is represented as a sequence of calls to three minimal APIs:

- `grasp(obj, euler)`: grasps the object with the specified orientation, implemented via sampling-based search over feasible grasp poses.
- `release()`: opens the gripper to let go of the object.
- `move(pos, euler)`: moves the gripper to a target position and orientation.

Our `grasp(obj, euler)` API is designed to robustly grasp objects given a desired gripper orientation, specified in Euler angles. It takes as input the target object and the intended orientation of the gripper during grasping. We then randomly sample pairs of farthest points on the object's surface to identify potential contact positions for grasping. These candidates are filtered based on their alignment with the input gripper orientation, discarding those that deviate significantly. For each valid candidate, we attempt to execute a grasp in simulation by lifting the object and checking whether it remains securely held. Upon the first successful attempt, the corresponding grasp pose is recorded. Subsequent calls to `grasp(obj, euler)` reuse this cached pose, ensuring consistent and efficient execution. `move(obj, euler)` is implemented by first solving inverse kinematics (IK) to get target joint pose, followed by motion planning to avoid collisions.

## 3.4 Optimizer

While LLMs and VLMs offer strong priors and can propose reasonable initial designs and action plans, they often lack the precision required for physically-grounded tasks. As demonstrated in Fig. 2 (b), the initial trajectory and design proposed by VLM agent fails to accurately execute the task. Therefore, after the initial selection of candidate tool shapes and trajectories, we perform a *joint*

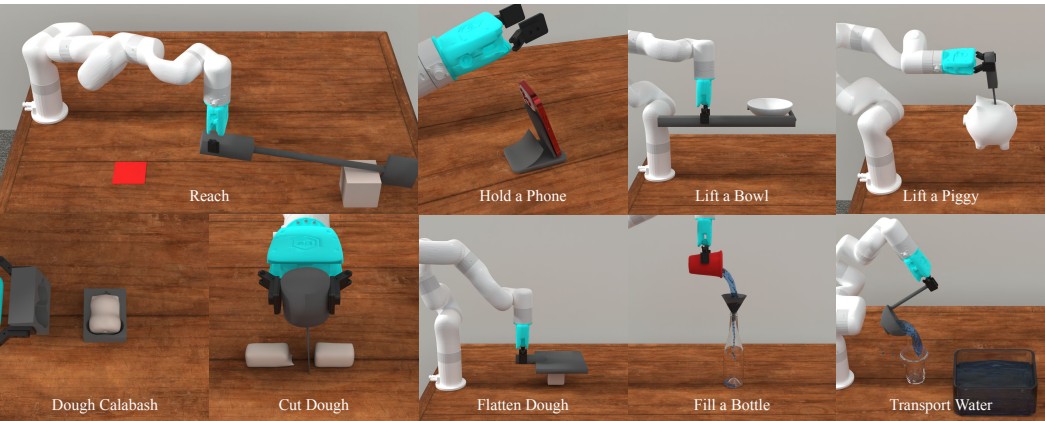

Figure 3: **Gallery of the curated robotic manipulation tasks**, that spans diverse physical and functional settings, including rigid, deformable, and fluid objects. This diversity allows comprehensive testing of tool design and usage capabilities across realistic and challenging scenarios.

*optimization* process to refine both components in tandem. This step is crucial, since a suboptimal pairing—even with a plausible tool or trajectory alone—can fail to complete the task.

In our pipeline, the agent in previous steps determines both *which parameters to optimize* and their *initial values*. Let $\mathbf{q} \in \mathbb{R}^{d_q}$ denote the trajectory parameters and $\mathbf{s} \in \mathbb{R}^{d_s}$ denote the tool shape parameters. As illustrated in Fig. 2 (c), $\mathbf{q}$ corresponds to the input parameters (position and Euler angles) of the `Move` API, while $\mathbf{s}$ defines the dimensions of the cylindrical and box components that make up the tool.

We support geometry optimization by allowing each shape parameter $s_i \in \mathbf{s}$ to vary within a range of $[0.5\,s_i^{(0)}, 2.0\,s_i^{(0)}]$, where $s_i^{(0)}$ is the initial value. The trajectory parameters $\mathbf{q}$ include translational and rotational waypoints in Cartesian space. Translational components are constrained to lie within $\pm 0.2$ m of their initial values, while rotational components may vary within $\pm \pi$ radians.

Since the initial proposal $(\mathbf{s}^{(0)}, \mathbf{q}^{(0)})$ generated by the language model often fails to complete the task, we apply the Covariance Matrix Adaptation Evolution Strategy (CMA-ES) [21] to optimize the joint parameter $(\mathbf{s}, \mathbf{q})$. CMA-ES searches for improved solutions by iteratively sampling a population of candidates $\{(\mathbf{s}_i, \mathbf{q}_i)\}_{i=1}^{\lambda}$, where $\lambda = 20$ is the population size. Each optimization run for $50$ iterations.

To evaluate each candidate solution, we execute the trajectory with the corresponding tool in the Genesis [4] simulator and compute a task-specific scalar objective: $\mathcal{M}(\mathbf{s}_i, \mathbf{q}_i)$, which reflects the success or quality of task completion. This objective is then used to guide CMA-ES toward better configurations. Please refer to Appendix C for the specific evaluation metrics used for each task.

## 4 Experiments

### 4.1 Setup

**Task Design**. As in Fig. 3, we curated 9 robotic manipulation tasks inspired by everyday human activities. These tasks vary widely in both physical properties and functional requirements, involving objects made of rigid, liquid, and soft bodies. This diversity ensures that our tool design module accommodates a range of physical interactions, while also allowing our pipeline to be rigorously evaluated across realistic and varied challenges encountered in daily life. The task details can be found in Appendix C.

**Metrics**. We scale each task's metric $\mathcal{M}$ to a standard score $P \in [0, 1]$. We run each experiment 8 times to report the best score $P_{best}$ and the overall success rate $SR = \frac{\sum \mathcal{I}[P > 0.8]}{8}$. Please refer to Appendix C for more details.

We test our pipeline using the Genesis [4] simulator and conduct experiments on both NVIDIA GeForce RTX 4090 GPUs and 2080 Ti GPUs.

Table 1: **Main results in simulation**. We report $P_{best}$ / SR, where $P_{best}$ is the best performance out of 8 trials (normalized to 0-1), and SR is the success rate over those 8 trials. A trial is considered successful if the P score exceeds 0.8.

| Method | Reach | Hold a Phone | Lift a Bowl | Lift a Piggy | Dough Calabash |
|---|---|---|---|---|---|
| No Tool | 0.00 / 0.0% | 0.09 / 0.0% | 0.33 / 0.0% | 0.00 / 0.0% | 0.56 / 0.0% |
| Retrieval | 0.09 / 0.0% | 0.84 / 25.0% | 0.30 / 0.0% | **1.00** / 25.0% | 0.21 / 0.0% |
| Meshy[37] | 0.69 / 0.0% | **0.99** / 67.5% | 0.13 / 0.0% | **1.00** / 25.0% | **0.93 / 25.0%** |
| Ours | **0.93 / 50.0%** | 0.95 / **75.0%** | **1.00 / 50.0%** | **1.00 / 87.5%** | 0.84 / **25.0%** |

| Method | Flatten Dough | Cut Dough | Fill a Bottle | Transport Water | Overall |
|---|---|---|---|---|---|
| No Tool | 0.24 / 0.0% | 0.82 / 25.0% | 0.08 / 0.0% | 0.00 / 0.0% | 0.24 / 2.8% |
| Retrieval | 0.56 / 0.0% | **1.00 / 37.5%** | 0.12 / 0.0% | 0.67 / 12.5% | 0.53 / 11.1% |
| Meshy[37] | **1.00** / 12.5% | 0.86 / **37.5%** | 0.85 / 25.0% | 0.00 / 0.0% | 0.72 / 21.4% |
| Ours | **1.00 / 62.5%** | 0.95 / **37.5%** | **0.87 / 37.5%** | **0.89 / 25.0%** | **0.94 / 50.0%** |

## 4.2  Baselines

We compare our method against several baseline approaches:

**No Tool**: The robot attempts to complete the task using only its default end-effector. An LLM agent is prompted to generate an initial manipulation trajectory, which is then optimized using CMA-ES.

**Retrieval**: Suitable tools are selected from a curated 3D asset database based on semantic similarity to the task description, where we use BlenderKit [6]'s embodied search engine to retrieve the top-matching tools by "a tool for $\mathcal{T}$". The LLM agent provides an initial trajectory, which is subsequently optimized.

**3D Generation (Meshy)**: We use Meshy [37], a pretrained text-to-3D generation model, to directly synthesize tool meshes from prompts such as "a tool for $\mathcal{T}$". As in other baselines, the trajectory is initialized by an LLM and refined using CMA-ES.

**3D Editing (ShapeTalk)**: We adopt ShapeTalk [1], a framework to facilitate fine-grained 3D shape editing through natural language instructions, as a baseline for collaborative 3D editing. It is integrated with the Critic agent to iteratively generate and refine tool designs via natural language interaction. Still, the trajectory is initialized using an LLM prompt and optimized with CMA-ES.

**Results** As shown in Tab. 1, our method consistently outperforms all baseline approaches in both task success rate and overall performance across most tasks.

Baseline *No Tool* struggles to complete any of the tasks, highlighting the essential role of purposeful tool design in enabling complex manipulations. In rare cases, such as *cutting dough*, success is possible, but often in unintended ways—for instance, the robot nearly crushes the dough, breaking it into just two uneven parts rather than executing a clean cut.

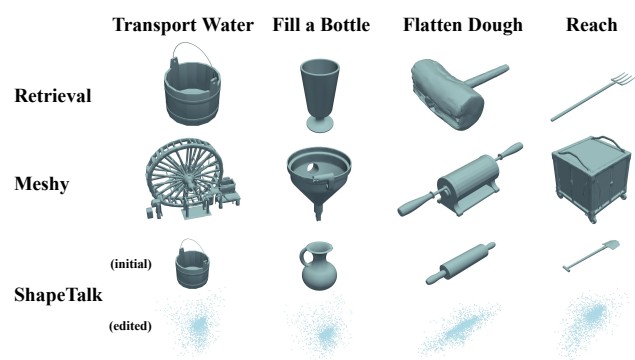

Figure 4: Failure cases of baseline methods.

Baseline *Tool Retrieval* offers moderate improvement but often selects tools originally designed for human use, such as a standard rolling pin for dough flattening, which are poorly suited for robotic grasping and control. Additionally, retrieval systems rely heavily on keyword-based matching and frequently misalign functional intent with suitable geometry. For example, as shown in Fig. 4, querying *flatten dough* may yield a hammer due to linguistic associations, despite the tool being physically inappropriate. Similarly, retrieval might suggest a fork for *reaching*, or return tools with mismatched scale and proportions.

For the *Meshy* baseline, due to their flexibility in shape generation and strong priors, they perform reasonably well on tasks that require specific tool geometries, such as *dough calabash, cut dough, fill bottle*, and *hold a phone*. These tasks benefit from the model's ability to generate complex and diverse shapes. However, this method tends to over-emphasize visual realism, such as smoothness, symmetry, and texture, rather than functional utility. As illustrated in Fig. 4, in the bottle-filling task, the model generates a funnel with three pipes, disrupting controlled water flow. These examples highlight a common limitation: despite visually plausible outputs, the lack of functional grounding often leads to failures in physical execution.

*ShapeTalk*, a representative 3D editing method, falls short in all our tasks. While it allows iterative editing, its reliance on natural language and implicit representations makes it difficult to perform meaningful modifications. In practice, we observe that edits often lead to incoherent or unusable tools. We deem that limitation stems from its lack of geometric and functional grounding. In contrast, our method leverages a parameterized and part-based representation with explicit assembly logic, making it inherently more editable, tunable, and interpretable. This design enables precise, semantically-meaningful modifications and better adaptation to task-specific constraints.

### 4.3 Tool Diversity

As shown in Fig. 5, our framework can generate diverse tool designs for the same task, each offering a different functionality, such as pushing, scooping, or enclosing. This variety arises from our generative process, which explores multiple valid solutions guided by task goals and physical constraints.

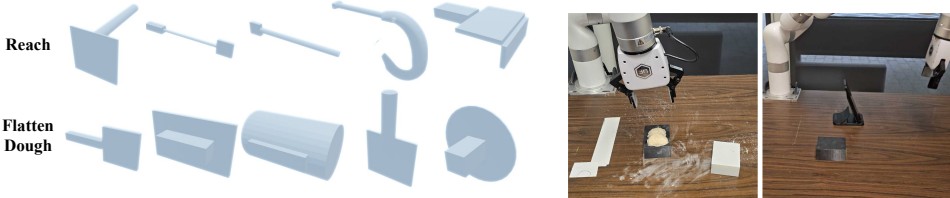

Figure 5: Diverse effective tools generated by our system for the *reach* task (top) and the *flatten dough* task (bottom).

Figure 6: Real-world execution of tools generated by our system for the *dough calabash* task (left) and *hold a phone* task (right).

### 4.4 Real-world Experiments

**Hardware setup**. To validate the real-world applicability of our method, we 3D printed the tools designed for the *hold a phone* and *dough calabash* tasks and tested them using a physical XArm7 robot equipped with a parallel gripper. As shown in Fig. 6, the generated tools are physically feasible, structurally stable, and effectively fulfill their intended functions. The robot was able to grasp, manipulate, and interact with the tools in real-world settings, demonstrating that our parameterized design and optimization pipeline produces artifacts suitable for real deployment.

**Long-horizon Task**. In addition, we conducted a long-horizon task—making a sesame pancake—to evaluate the system's performance across multiple, sequential manipulation steps. As shown in Fig. 1 (top row), the system autonomously designed and used distinct tools for each subtask, including flattening the dough, scooping sauce, spreading it evenly, and sprinkling sesame seeds. This demonstrates the effectiveness of our method in decomposing complex tasks and its ability to generate diverse, task-specific tools and coordinated usage plans that work in combination to achieve multi-step goals.

### 4.5 Ablation Study

We conduct ablation studies to assess the contribution of each major component of our pipeline. Specifically, we remove:
- the text-to-3D generator, replacing generated components with primitives only;
- the trajectory optimizer, using the initial action plan from the Designer without refinement;
- the tool optimizer, keeping the initial tool geometry fixed during execution.

Table 2: **Results of ablation study**: Dropping any component degrades performance, with optimization emerging as the most critical contributor.

| Method | Best Performance ↑ | Success Rate ↑ |
|---|---|---|
| Ours w/o Text-to-3D | 0.69 | 33% |
| Ours w/o Optimizer | 0.28 | 5% |
| Ours w/o Tool Opt | 0.67 | 32% |
| Ours | **0.94** | **50%** |

These ablations help isolate the impact of generative modeling, action refinement, and geometry tuning, and confirm that each component plays a critical role in overall task success and physical feasibility.

As shown in Tab. 2, removing the text-to-3D generator and relying solely on primitive shapes leads to a noticeable drop in both best performance and success rate, underscoring the importance of generative modeling in producing task-appropriate components. This is particularly evident in tasks like *fill a bottle* and *hold a phone*, where simple primitives fail to capture the necessary geometry, while text-to-3D offers the flexibility to generate complex and functional designs. Disabling the trajectory optimizer—thereby skipping action refinement—results in the most significant performance drop (from 0.94 and 50% success rate to 0.28 and 5%), demonstrating that relying solely on the VLM agent to generate trajectories is insufficient for accurate task execution. This highlights the critical role of the trajectory optimization phase. Additionally, removing the joint optimization of tool geometry leads to reduced performance. Although the tools may appear visually correct and can pass the Critic agent's evaluation, certain tasks—such as *Fill a Bottle* and *Lift a Piggy*—require high geometric precision for success. In such cases, jointly optimizing tool geometry is essential for achieving reliable performance. Together, these results demonstrate that each component plays a vital role in the robustness and effectiveness of the overall pipeline.

### 4.6 Failure Analysis

While our method achieves strong performance overall, we observe several failure cases arising from different stages of the pipeline. Design failures often stem from misalignment between the Designer agent and the Meshy text-to-3D generator. The Designer may specify only a single functional part, relying on Meshy to generate geometry, but Meshy's outputs are not always controllable in terms of size, orientation, or detail. As a result, the generated mesh may not match the designer's intent, leading to tools that are difficult to grasp or use effectively. Tool use failures typically occur when the tool's orientation is poorly defined. Designers often provide ambiguous axes or overly generic rotations (e.g., 90° or 180° flips), which are not sufficient for nuanced interactions like scooping or precise alignment, complicating execution and downstream optimization. Grasping failures arise from the limitations of our grasp API, especially when the tool is too heavy or when the required motion is abrupt, causing the grasp to fail during simulation. Finally, optimization failures occur in tasks involving tools with many shape parameters or long complex trajectories.

## 5 Conclusion

We present **RobotSmith**, a unified framework that enables fully autonomous tool design and use for robotic manipulation by combining the semantic reasoning capabilities of large foundation models with physics-based optimization. Extensive experiments across a variety of challenging manipulation tasks, both in simulation and the real world, demonstrate the effectiveness, robustness, and practicality of our approach. We believe this work represents a step toward more generalizable, embodied intelligence capable of solving long-horizon tasks through physical tool creation and interaction, and paves the way for future intelligent manufacturing and adaptive robotic assistants.

**Limitation**. While our framework introduces a modular and expressive tool representation that supports both procedural and generative components, the current optimization process only involves scaling, rotation, and translation. This may restrict exploring more complex structural changes that could better adapt tools to specific tasks. Enabling richer forms of tool modification, such as topology editing or part reconfiguration, could further improve their diversity, functionality, and effectiveness.

**Acknowledgement** This project was supported by MURI N000142412748, NSF IIS-2404386 and NVIDIA.

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

# A Tool Design

## A.1 An example of tool representation

```
{
  "name": "Funnel Tool with Handle",
  "parts": [
    {
      "geom": "mesh",
      "prompt": "metal funnel for liquid pouring",
      "parameters": [0.3, 0.3, 0.15],
      "is_graspable": false
    },
    {
      "geom": "cylinder",
      "prompt": "",
      "parameters": [0.02, 0.15],
      "is_graspable": true
    }
  ],
  "assembly": "def assembly(parts: list[dict]):    ...",
  "placement": "def placement():    ... # Place the tool"
}
```

Each part in the tool representation specifies a geometric component using the `"geom"` field, which supports six options: `"mesh"`, `"cylinder"`, `"cube"`, `"ball"`, `"ring"`, and `"tube"`. The `"is_graspable"` field indicates whether the robot gripper can grasp the part. To ensure a stable and fixed grasping pose, only one part should be marked as graspable. The interpretation of the `"parameters"` field depends on the selected geometry type:

- `"mesh"`: This part is a mesh generated by a text-to-3D model using a prompt. The parameters `[lx, ly, lz]` represent the dimensions of the bounding box after scaling, corresponding to the lengths along the X, Y, and Z axes.
- `"cube"`: The part is a cube. The parameters `[sx, sy, sz]` specify the scale along the X, Y, and Z axes.
- `"ball"`: The part is a ball. The parameter is a single value `[radius]`, representing the radius of the sphere.
- `"cylinder"`: The part is a cylinder. The parameters `[radius, height]` define the radius of the circular base and the height along the Z-axis.
- `"ring"`: The part is a ring. The parameters `[radius, thickness]` describe a flat torus, where `radius` is the major radius and `thickness` is the cross-sectional thickness.
- `"tube"`: The part is a tube. The parameters `[radius, length, thickness]` define a hollow cylindrical tube with a given outer radius, length, and wall thickness.

The `assembly` function takes the list of parts as input and uses several editing APIs to merge them into a complete tool. An example assembly function is shown below:

```
def assembly(parts: list[dict]):
    # Import necessary modules from trimesh
    import trimesh

    # Part 1: Create the funnel using text-to-3D generation
    funnel_part = parts[0]
    # Generate a funnel mesh from the prompt
    funnel = generate_3d(funnel_part["prompt"])

    # Align the funnel so that its bounding box is centered and axis-
        aligned
    funnel = rotate_to_align(funnel)
```

```python
# Get the current bounding box dimensions
bbox = get_axis_aligned_bounding_box(funnel)
dims = [bbox[1] - bbox[0], bbox[3] - bbox[2], bbox[5] - bbox[4]]

# We want the funnel's x-dimension (largest by our convention) to
    match the provided parameter
target_dims = funnel_part["parameters"]
# Compute a uniform scale factor based on x-dimension
scale_ratio = target_dims[0] / dims[0] if dims[0] != 0 else 1.0
funnel = rescale(funnel, scale_ratio)

# Part 2: Create the handle using a primitive cylinder
handle_part = parts[1]
handle = primitive("cylinder", handle_part["parameters"])
handle = rotate_to_align(handle)

# Position the handle so that it attaches to the side of the
    funnel
# We assume the funnel is centered at the origin, so we shift the
    handle left along x
handle = move(handle, (-dims[0], 0, 0))

# Optionally, we could adjust the vertical offset of the handle if
    needed, e.g., move slightly down
# handle = move(handle, (0, 0, -dims[2]))

# Combine the two meshes into one tool mesh
tool_mesh = trimesh.util.concatenate([funnel, handle])

# Export the assembled tool to an OBJ file
tool_mesh.export('tool.obj')

# Return the list of exported asset filenames
return ['tool.obj']
```

## A.2 An example of tool designer

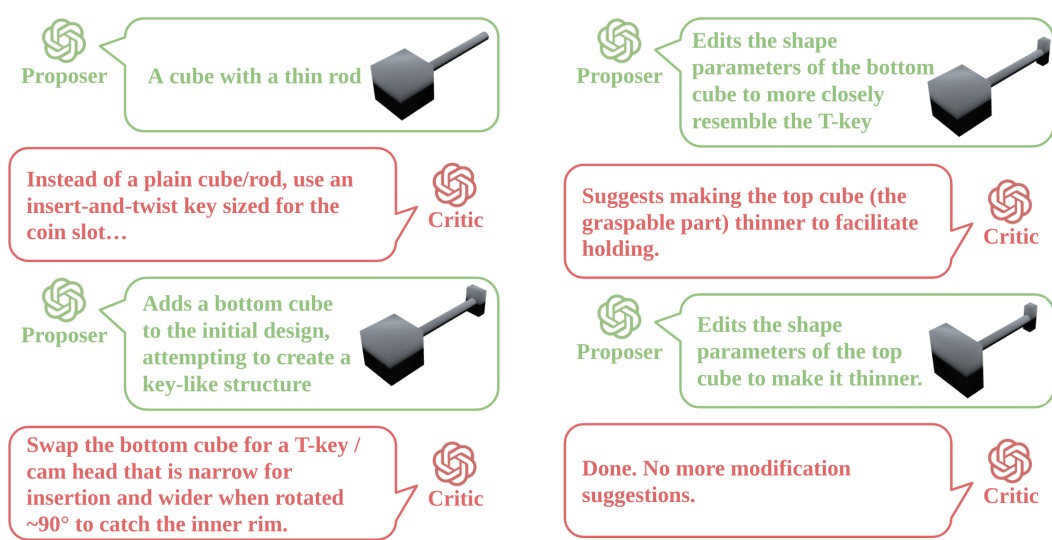

Figure 7: A 4-round conversation between Proposer and Critic for the Piggy task.

# B   APIs and Prompt Templates

## B.1   Proposer

### B.1.1   Proposer Prompt Template

```
Now you have a robotic task:

On the table, there is a robot arm with a parallel gripper. The task
    goal is to $TASK_DESCRIPTION$. For more detailed scene setup, here
     is a code:

$3D_CONFIGURATION$

You need to design a tool to help finish the task.

You have 2 APIs to generate 3D meshes:

[DESCRIPTIONS OF GENERATION APIS]

You have several APIs for 3D editing:

[DESCRIPTIONS OF MODIFICATION APIS]

To describe a generated tool, you will use a json format. Here is an
    example:
```json
{
    "name": a string,
    "parts": [
        {
            "geom": one in ['cube', 'ball', 'cylinder', 'ring', 'tube
                ', 'mesh'],
            "prompt": if "geom" is 'mesh', there is a string
                representing a text prompt for text-to-3d generation
                model; otherwise is an empty string,
            "parameters": if "geom" is 'mesh', the parameters are (lx,
                 ly, lz), the length in x/y/z directions;
                          if not, the parameters should be the same as `
                              primitive_scale` in primitive().
            "is_graspable": True/False, if it is True, then robot arm
                can grasp this part, otherwise the robot arm cannot
                grasp.
        },
        {
            ...
        },
    ],
    "assembly": a complete python code to assemble all the parts above
         into the whole tool using 3D editing APIs provided.
        """
            def assembly(parts: list[dict]):
                ...
                # the input parameter parts here is a list of
                    dictionaries, you need to call primitive() or
                    generate_3d() manually with parameters in parts.
                # in the function, the above 3D editing APIs are used
                    to assemble each part into the whole tool.
                # the function should export 3D assets of the tool
                    into one or more obj files, each asset should be
                    on the x-y plane.
                # the function returns a list of file names of the
                    exported 3D assets.
        """
```

```
    "placement": a complete python code to place the tool in the scene
        .
        """
            def placement():
                self.tool = scene.add_entity(
                    gs.Morph.Mesh(
                        filename="tool.obj",
                        pos=(0.0,0.0,0.0),
                        euler=(0,0,0),
                        scale=(1,1,1),
                    )
                )
                # the filename here should match that saved in
                    assemble_func, and the position, rotation and
                    scale should be approprate to be here.
        """
}
```

You should make sure:
The tool is saved. All the 3D editing functions in assembly func have
    return values, which should be received.
$USER_PROVIDED_PRINCIPLES$
```

### B.1.2 Generation APIs

The following 2 APIs are available for the Proposer to create 3D geometry:

```python
def primitive(primitive_name, primitive_scale)
    """
    Create a 3D primitive mesh, centered at the origin.
    Args:
        primitive_name (str): One of {'cube', 'ball', 'cylinder', '
            ring', 'tube'}.
        primitive_scale (list of float):
            - If cube, [sx, sy, sz]
            - If ball, [radius]
            - If cylinder, [radius, height]
            - If ring, [radius, thickness]
            - If tube, [radius, length, thickness]
    Returns:
        trimesh.Trimesh: The resulting mesh, centered at (0, 0, 0).
    """

def generate_3d(name):
    """
    Call a pretrained text-to-3D model to generate a single solid 3D
        object.
    Args:
        name (str): the text prompt to generate 3D shape. Should be
            very simple, like several words.
    Returns:
        trimesh.Trimesh: The resulting mesh, at a random position and
            of a random size.
    Tips:
        Remember to rescale and align the generated mesh, since the
            size and position would be random.
    """
```

### B.1.3 3D Editing APIs

The following 11 APIs are available for the Proposer to modify 3D geometry:

```python
def rotate_to_align(mesh1):
    """
    Rotate and translate the mesh so that:
    - Its bounding box is centered at the origin (0,0,0).
    - The largest dimension of the bounding box aligns with +X,
    - The second-largest aligns with +Y,
    - The smallest aligns with +Z.
    The mesh remains axis-aligned and is returned in-place.
    Args:
        mesh (trimesh.Trimesh): The input mesh (modified in-place).
    Returns:
        trimesh.Trimesh: The rotated+translated mesh, for convenience.
    """

def get_position(mesh):
    """
    Return the position of the mesh as (x, y, z).
    Args:
        mesh (trimesh.Trimesh): The mesh.
    Returns:
        tuple: (x, y, z)
    """

def get_axis_align_bounding_box(mesh):
    """
    Return the axis-aligned bounding box of the mesh as (min_x, min_y,
        min_z, max_x, max_y, max_z).
    Args:
        mesh (trimesh.Trimesh): The mesh.
    Returns:
        tuple: (min_x, min_y, min_z, max_x, max_y, max_z)
    """

def get_volume(mesh):
    """
    Return the volume of the mesh in cubic units.
    Args:
        mesh (trimesh.Trimesh): A closed, manifold mesh.
    Returns:
        float: The volume of the mesh.
    """

def rescale(mesh, ratio):
    """
    Uniformly scale the mesh by the given ratio about the origin (0,
        0, 0).
    Args:
        mesh (trimesh.Trimesh): The mesh to scale (modified in-place).
        ratio (float): The scale factor to apply.
    Returns:
        trimesh.Trimesh: The scaled mesh (for convenience).
    """

def move(mesh, offset):
    """
    Translate the mesh by the given offset (x, y, z).
    Args:
        mesh (trimesh.Trimesh): The mesh to move in-place.
        offset (tuple or list of float): The translation offset (dx,
            dy, dz).
    Returns:
        trimesh.Trimesh: The transformed mesh (for convenience).
    """

def empty_grid():
```

```
    """
    Create an empty 256x256x256 boolean occupancy grid from -0.5 to
        +0.5 in each axis.
    Returns:
        dict: A dictionary containing:
            - 'data': np.ndarray of shape (256, 256, 256), dtype=bool
                (all False initially).
            - 'res':  integer (256).
            - 'min_bound': np.array([-0.5, -0.5, -0.5]).
            - 'max_bound': np.array([0.5, 0.5, 0.5]).
    """

def add_mesh(grid, mesh):
    """
    Convert 'mesh' into a volume of occupied voxels using an SDF (
        signed-distance) test,
    then mark those voxels as True in 'grid'.
    Args:
        grid (dict): The grid dictionary from empty_grid().
        mesh (trimesh.Trimesh): A triangular mesh (assumed to fit in
            [-0.5, 0.5]^3).
    Returns:
        dict: The updated grid, same reference as input.
    """

def sub_mesh(grid, mesh):
    """
    Convert 'mesh' into a volume using an SDF, then set those voxels
        to False
    (subtract from the grid).
    """

def cut_grid(grid):
    """
    Return two new grids of the same resolution (256x256x256):
    - grid_bottom: occupies only z < 0
    - grid_up: occupies only z >= 0
    by zeroing out the complementary region in each grid.
    Each returned grid has:
    - 'data': a (256,256,256) boolean array
    - same 'min_bound' and 'max_bound' as the original
    - same 'res' as the original
    Args:
        grid (dict): Must have keys 'data', 'res', 'min_bound', '
            max_bound'.
            'data' is a 3D boolean array: (256,256,256).
    Returns:
        (dict, dict): (grid_up, grid_bottom)
    """

def grid_to_mesh(grid, do_simplify=True, target_num_faces=3000):
    """
    Convert a 3D occupancy grid into a surface mesh using Marching
        Cubes.
    Optionally simplify the mesh using Open3D's quadric decimation.
    Args:
        grid (dict): A dictionary with keys:
            - 'data': (256,256,256) boolean array (True = occupied).
            - 'res': int, resolution (e.g. 256).
            - 'min_bound': np.array([x_min, y_min, z_min]).
            - 'max_bound': np.array([x_max, y_max, z_max]).
        do_simplify (bool): Whether to perform mesh simplification (
            default True).
        target_num_faces (int): If simplifying, the target number of
            faces.
```

```
    Returns:
        trimesh.Trimesh: The extracted (and optionally simplified)
            mesh.
            If the grid is empty or no surface is found, faces might
                be empty.
    """
```

## B.2 Critic Prompt Template

Here is the prompt template for the Critic agent:

```
Now you have a robotic task:

On the table, there is a robot arm with a parallel gripper. The task
    goal is to $TASK_DESCRIPTION$. For a more detailed scene setup,
    here is a code:

$3D_CONFIGURATION$

Here are the multi-view figures of a tool designed for the task. Does
    that look good? If so, reply DONE; if not, give modification
    suggestions.

Here are some important aspects for you to examine if the tool looks
    good:
1. A good tool should be easy to grasp.
2. A good tool should be in good shape, and each part looks connected.
$USER_PROVIDED_PRINCIPLES$
[RENDERED_FIGURES]
```

## B.3 Tool User Prompt Template

```
Now you have a robotic task:

On the table, there is a robot arm with a parallel gripper. The task
    goal is to $TASK_DESCRIPTION$. You have a tool designed for the
    task. For more detailed scene setup, here is the scene setup code
    and a rendered image:

$3D_CONFIGURATION$

Now you need to provide a robotic manipulation trajectory using the
    following 3 APIs:

```python
move(pos, euler) # move the gripper to the target position and
    orientation
grasp(obj, euler) # auto calculate a grasping pose, aligning the given
     gripper orientation, to steadily grasp obj
release() # open the gripper to the widest
```

Please write a manipulate() function with the 3 APIs provided above.
    Note you can only move tools or rigid objects in the scene.
```

# C  Task Details

## C.1  Dough Calabash

**Task Description** $\mathcal{T}$: "make the dough into a calabash liked shape"

**Initial Scene Configuration** $\mathcal{S}_0$: `dough=Box(size, pos)` # a cubic dough

**Metric** $\mathcal{M}$: `min(1, 2 * CLIP_SCORE(rendered_dough_image, "calabash-shaped dough"))`

## C.2 Reach

**Task Description** $\mathcal{T}$:`"reaching the cube outside workspace"`

**Initial Scene Configuration** $\mathcal{S}_0$: `cube=Box(size0, pos0)` # a cube out of robot arm's reachability

**Metric** $\mathcal{M}$: `max(0, (cube.pos-target_pos).norm()/(pos0-target_pos).norm())`

## C.3 Flatten Dough

**Task Description** $\mathcal{T}$:`"flatten the dough to a height smaller than 0.03"`

**Initial Scene Configuration** $\mathcal{S}_0$: `dough=Box(size0, pos0)` # a cubic dough

**Metric** $\mathcal{M}$: `1 - max(0, (dough.height-0.03)/(height0-0.03))`

## C.4 Hold a Phone

**Task Description** $\mathcal{T}$:`"keep the phone face upfront rather than lying on the table"`

**Initial Scene Configuration** $\mathcal{S}_0$: `phone=Mesh('iphone.obj', pos, euler))` # a phone

**Metric** $\mathcal{M}$: `1 - (90-phone.euler[0])/90)`

## C.5 Fill Bottle

**Task Description** $\mathcal{T}$:`"pour the cup of water into the bottle"`

**Initial Scene Configuration** $\mathcal{S}_0$:

`bottle=Mesh('bottle.obj', pos, euler)` # a bottle

`cup=Mesh('cup.obj', pos, euler)` # a cup

`water=Cylinder(radius, height, pos))` # a cup of water

**Metric** $\mathcal{M}$: `1 - max(0, (dough.height-0.03)/(height0-0.03))`

## C.6 Lift a Bowl

**Task Description** $\mathcal{T}$:`"lift up the large bowl without touching inner surface"`

**Initial Scene Configuration** $\mathcal{S}_0$: `bowl=Mesh('bowl.obj', pos, euler)` # a large bowl

**Metric** $\mathcal{M}$: `1 - max(0, (0.1-bowl.pos[2])/0.1)`

## C.7 Cut Dough

**Task Description** $\mathcal{T}$:`"cut the dough into two even parts"`

**Initial Scene Configuration** $\mathcal{S}_0$: `dough=Box(size0, pos0)` # a cubic dough

**Metric** $\mathcal{M}$:

`dough1, dough2 = KMeans(n_cluster=2).fit(dough)`

`1 - max(0, (0.2-(dough1.pos-dough2.pos).norm())/0.2`

## C.8 Transport Water

**Task Description** $\mathcal{T}$:`"transport as more as possible water into the cup from the tank"`

**Initial Scene Configuration** $\mathcal{S}_0$:

$\mathcal{S}_0$: `bowl=Mesh('tank.obj', pos0, euler0)` # a a water tank

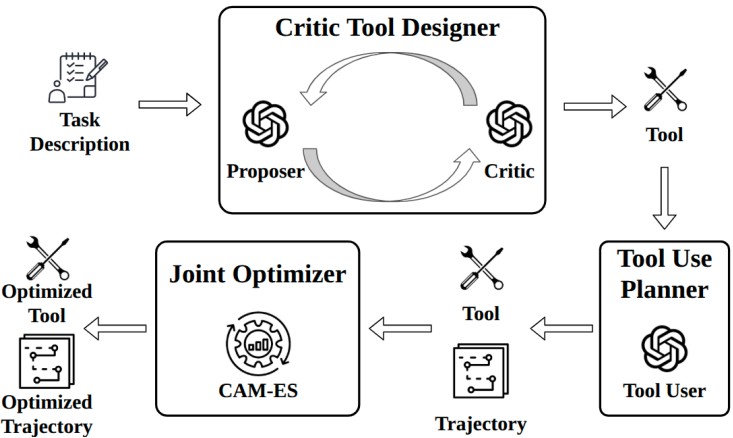

Figure 8: A brief pipeline figure to explain the workflow.

**Initial Scene Configuration** $\mathcal{S}_0$: `cup=Mesh('cup.obj', pos1, euler1) # an empty cup`
`water=Box(size0, pos0) # a tank of water`
**Metric** $\mathcal{M}$: `water_in_cup.volumn / cup.volumn`

## C.9 Lift a Piggy

**Task Description** $\mathcal{T}$:`"lift up the piggy bank"`
**Initial Scene Configuration** $\mathcal{S}_0$:
`piggy=Mesh('piggy.obj', pos0, euler0) # a piggy bank with insertion hole on top of it`
**Metric** $\mathcal{M}$: `1 - max(0, (0.3-piggy.height)/0.3)`

# D  Societal Impact

**Positive Impact**. Our work contributes toward more capable and autonomous robotic systems by enabling them to design and use tools tailored to specific tasks. This has the potential to greatly benefit sectors such as elder care, disaster response, and manufacturing, where adaptive, general-purpose robots could reduce human labor in hazardous or repetitive environments. By making tool-use more accessible and efficient, our framework could also accelerate innovation in robotics education and research, fostering creativity and reducing the need for manual engineering. Moreover, the open-sourcing of our APIs and representations promotes transparency and reproducibility, encouraging responsible development within the research community.

**Negative Impacts**. As with many advances in automation and robotics, our system could contribute to labor displacement in industries reliant on manual or semi-skilled tasks, particularly in basic manufacturing. Additionally, the misuse of autonomous tool design—for example, in adversarial contexts—raises concerns. Our system's reliance on large foundation models also inherits potential biases or inaccuracies present in those models. Careful deployment, human oversight, and continuous auditing are necessary to mitigate these risks and ensure that such systems are used for socially beneficial purposes.

# E  Compute Resources

Our experiments involve two main components: querying large pretrained models and running optimization in simulation. We use API calls to models such as O3-Mini and Meshy for tool design and initial tool use planning. These API-based components do not place any demand on local compute resources. For simulation and optimization, we run CMA-ES within the Genesis simulator, which supports both CPU-only and GPU-accelerated platforms. Our tasks do not require large GPU memory,

and many experiments were successfully conducted on a single NVIDIA 2080 Ti GPU, and others were conducted on NVIDIA GeForce RTX 4090 GPUs. While more powerful GPUs can improve simulation efficiency, they are not essential for reproducing our results.

