# OpenReview forum: "RobotSmith: Generative Robotic Tool Design for Acquisition of Complex Manipulation Skills"
_NeurIPS.cc/2025/Conference — NeurIPS 2025 poster_

### Official Review · Reviewer_BMgf · 2025-06-13

**Clarity:** 3
**Significance:** 3
**Originality:** 3
**Rating:** 4
**Confidence:** 4

**Summary:**

This paper introduces RobotSmith, a framework to design specialized tools for diverse robot tasks using foundation models and physics simulation. RobotSmith has three steps: (1) iterative tool design, where a proposer VLM creates tools via a specialized programmatic representation and a critic VLM provides feedback based on renderings of the tool; (2) planning, where another VLM calls high-level robot APIs to synthesize a trajectory; and (3) tool-trajectory joint optimization, where an evolutionary strategy refines the VLM's parameters by evaluating candidates in simulation with a task objective function. The authors validate their method on 9 diverse manipulation tasks, including those involving soft bodies and liquids, and show that RobotSmith outperforms baselines: no tool use, human tool retrieval, and mesh generation. Additionally, generated tools are 3D printed and applied to 2 tasks in the real world, demonstrating successful transfer.

**Questions:**

Questions
1. The discussion on failure cases in Section 4.6 is very helpful, and I wonder if you could also include percentages for each type of failure. That is, which of these failure cases are most important to address for future work?
2. How does your tool representation compare with MJCF or URDF? I understand a key distinction is the ability to generate geometries, and are there other key advantages? I wonder if these commonly-used formats could be helpful for VLM generation, since they already exist in the internet training data.
3. Can you provide some more technical details on the main changes that allow RobotSmith to run in real? For instance, with the grasp API, how do you get the location of the object without ground truth simulator state?
4. Following Weakness 1, can you provide more details on the evaluation procedure and success rate of the real-world trajectories?

**Ethical Concerns:**

["NO or VERY MINOR ethics concerns only"]

**Final Justification:**

The authors addressed my questions and limitations in their rebuttal. Explanations and justifications were clear and helped me better understand their design. The work is overall convincing and solid, and the only main weakness is the real-world results. However, the authors do demonstrate that their generated tools are kinematically and geometrically feasible in real (given privileged state), and as real-world policy synthesis is not the focus of this work, I believe the real-world experiments are sufficient.

**Limitations:**

Yes

**Quality:**

3

**Strengths And Weaknesses:**

Strengths
1. The experiments show that RobotSmith's custom-designed tool representation is quite general-purpose, as it allows the VLM to generate many diverse geometries. The tasks that these tools complete is also very different, each requiring different geometric features.
2. Some of the generated results are validated in real, demonstrating potential for scalable real-world tool design. The long-horizon pancake-making task is especially impressive.
3. The combination of VLM generation with subsequent optimization is novel and addresses a key issue in VLMs lacking physical precision.
4. This paper tackles the difficult and promising challenge of tool-design, which has the potential to significantly expand a robot's capabilities.
5. The paper is written clearly, and the figures are overall illustrative.

Weaknesses
1. While the method is validated in real, it isn't evaluated over multiple trials. Section 4.4 seems to suggest the existence of successes in real, but does not elaborate on its effective success rate.
2. The planner only uses three API functions (grasp, release, move), which seem limiting and may restrict it from more freeform trajectories (eg, wiping). The grasp API is also called on the entire object, and the grasp point is not exact; I wonder whether this may hinder the use of a tool due to the heuristic grasp selection choosing an incorrect location.
3. Figure 2 seems a bit cluttered due to the many textual examples. It may be clearer to have two separate figures, an overview that more abstractly visualizes the method and another that shows examples in each step.

---

> ### Author Rebuttal · Authors · 2025-07-31
>
> *We appreciate the positive and insightful comments from you! We adress your concerns in details below.*
>
> **Weaknesses**
>
> > **1. While the method is validated in real, it isn't evaluated over multiple trials. Section 4.4 seems to suggest the existence of successes in real, but does not elaborate on its effective success rate. & Q4: Following Weakness 1, can you provide more details on the evaluation procedure and success rate of the real-world trajectories?**
>
> In high level, our pipeline is designed to work in simulation to generate tool-use data, which would be used to train a policy that can be deployed to the real-world in the future.
>
> Thus, our real-world experiments aim to validate that tools designed and trajectories optimized in simulation are structurally stable and physically feasible. Thus, we first build a simulated digital twin of the setup, perform tool design and trajectory optimization in simulation, and then 3D-print the tool and execute a selected successful trajectory in the real world.
>
> Although evaluating real-world robot performance isn't our main focus, we report the following success rates: *Dough calabash*: 9/20, *Hold a phone*: 20/20 (simple trajectory replay), *Pancake-making*: 1/4 (limited trials due to tedious scene resets).
>
> Despite that theoretically it should be all success, failures occur mainly due to:
>
> - **Shape variation**: Manual resets of deformable objects cause inconsistencies.
>
> - **Material changes**: Contact with water or dough (flour) alters surface friction, affecting grasp reliability.
>
>
> > **2. The planner only uses three API functions (grasp, release, move), which seem limiting and may restrict it from more freeform trajectories (eg, wiping). The grasp API is also called on the entire object, and the grasp point is not exact; I wonder whether this may hinder the use of a tool due to the heuristic grasp selection choosing an incorrect location.**
>
> The *Grasp*, *Release*, and *Move* APIs are a deliberately minimal set of high-level commands. Their purpose is to let the VLM provide semantic priors for tool use and action planning, which are then refined into accurate trajectories by downstream optimization. Wiping can be approximated by a sequence of *Move* calls. While we could make *Move* more free-form (e.g., via RL), that would introduce many more parameters and significantly increase optimization time. Our design therefore trades some expressiveness for computational efficiency.
>
> For grasp location, we want to clarify that sampling is not performed over the entire tool surface; it is restricted to parts explicitly designed for grasping (the *is\_handle* label in Fig. 2 of the main paper). By design, each tool should include a functionally graspable part. As a result, grasps are semantically appropriate and unlikely to interfere with the tool’s function.
>
>
>
> > **3. Figure 2 seems a bit cluttered due to the many textual examples. It may be clearer to have two separate figures, an overview that more abstractly visualizes the method and another that shows examples in each step.**
>
> We appreciate your advice regarding the clarity of Figure 2. We'll address this in the final version of the paper by adding a clean and brief figure that solely visualizes the overall pipeline more abstractly. The detailed examples, which currently contribute to the clutter, will then be separated into a distinct, dedicated figure.
>
>
> **Questions**
>
>
> > **1. The discussion on failure cases in Section 4.6 is very helpful, and I wonder if you could also include percentages for each type of failure. That is, which of these failure cases are most important to address for future work?**
>
> Thanks for finding our discussion on failure cases helpful and for the excellent suggestion to include failure percentages. While our method aims to address the limitations of baseline approaches (such as their failure to retrieve appropriate tools or realize simple geometries for certain function, as seen in Figure 4), we have conducted a further analysis, and the failures in our pipeline can be broken down as follows:
>
> - Invalid Code Generation (32%): A significant portion of failures occurs when the Proposer generates invalid code (e.g., code missing return value captions). Although the Critic detects failures, the Proposer is not always able to correct itself within the limited number of back-and-forth rounds.
>
> - VLM Multi-modal Understanding (47%): The largest source of failure is from the limited multi-modal understanding ability of VLMs. This can lead to undetected failures, such as the generation of non-functional or unreasonable shapes. We believe this will be a primary area for future work and is likely to be mitigated as VLMs continue to improve.
>
> - Optimization Limitations (15%): A smaller portion of failures can be attributed to the limited computational budget for the optimization process, which may not always find the optimal parameters.
>
> - Other Cases (6%): The remaining failures are due to various other miscellaneous issues.
>
> > **2. How does your tool representation compare with MJCF or URDF? I understand a key distinction is the ability to generate geometries, and are there other key advantages? I wonder if these commonly-used formats could be helpful for VLM generation, since they already exist in the internet training data.**
>
> Thanks for the excellent question that highlights a key design choice in our work. MJCF and URDF are highly structured formats designed to describe the kinematics and dynamics of articulated, rigid-body systems, such as robot arms. Our tool representation, in contrast, is designed to be highly expressive for generating novel, flexible tools with diverse geometries.
>
> Our tool representation is inspired by Constructive Solid Geometry (CSG), a well-established and powerful method widely used in CAD. This provides a key advantage in expressiveness. For example, it easily supports operations like set minus to create intricate concave shapes, which would be difficult to describe with the joint and link primitives of MJCF or URDF.
>
> It's a good point that presence in internet training data could be helpful. We believe that given CSG's popularity as a design choice, VLMs also have a strong understanding of this representation. Therefore, our approach combines the VLM's high-level generative capabilities with our representation that is a better fit for creating complex, task-specific geometries.
>
>
> > **3.Can you provide some more technical details on the main changes that allow RobotSmith to run in real? For instance, with the grasp API, how do you get the location of the object without ground truth simulator state?**
>
> As noted above, we execute an open-loop trajectory optimized in simulation just to prove the physics feasibility. The simulation provides ground-truth state information. To build the digital twin, we manually measure object positions relative to the robot arm—keeping real-to-sim transfer simple, as it's not our primary focus.
>
> *We hope that the provided new experiments and additional explanations have convinced you of the merits of our work. Please do not hesitate to contact us if you have other concerns.*
>
> *We appreciate your time! Thank you so much!*
>
> Best,
>
> Authors

---

> > ### Comment · Reviewer_BMgf · 2025-08-04
> >
> > Dear authors, thank you for your thoughtful response and new experiments. My questions and limitations have been clearly addressed, and while the real-world results are open-loop replays using digital twins, they do show that the method has potential in real.

---

### Official Review · Reviewer_Kk5s · 2025-06-28

**Clarity:** 2
**Significance:** 2
**Originality:** 2
**Rating:** 4
**Confidence:** 2

**Summary:**

The paper proposes a tool augmented VLLM framework, that designs tools for robotic agents as well as the trajectory for their use. Several components based on GPT3omini are used with a CMA-ES at the end to address LLM geometrical reasoning limitations.

**Questions:**

The main issues identified are with the clarity of the method and result interpretation.
- What components are trainable and what components are shared?
- What is the optimisation process?
- Clarifincation on evaluation is needed

**Ethical Concerns:**

["NO or VERY MINOR ethics concerns only"]

**Final Justification:**

There are some novelty concerns post rebuttal, specifically as the method is proposing in practice a tool-augmented MLLM in a new domain (the authors themselves mention several in the related work and the rebuttal). However, the results are convincing and the motivation for using MLLMs in the task is very clear and intuitive.

**Limitations:**

yes

**Paper Formatting Concerns:**

No formatting concerns

**Quality:**

3

**Strengths And Weaknesses:**

The paper is well written, explains the problem well and provides a solution.
However, there are several severe issues related to the clarity of the method.

- It is unclear what components are trainable if any are at all and how they are trained. This needs to be made explicit.
- Do the Proposed, Critic and Tool Planner share weights?
- The tool user is not mentioned in the problem setting/method overview.
- What are the APIs mentioned in 3.3? how are these obtained? are they open source libraries? these need better explanation.
- Optimisation process is not explained at all in the joint optimiser. This needs to be significantly more elaborate. Preliminaries for the CMA-ES should be added to the main paper or appendix as it is an important component of the pipeline.
- Table 1 results appear impressive, but some further investigation is necessary. What dataset is used? is this publicly available and if not, will it be made publicly available?
- The best and average of 8 runs is used, however these results are not very meaningful as they are (best could be an outlier). It would be much better to show mean +- std and since multiple runs are computed, statistical significance of results would be in order.
- This is the same for the ablation study.

---

> ### Author Rebuttal · Authors · 2025-07-31
>
> *We sincerely appreciate your detailed and constructive feedback. We have carefully considered your suggestions and enhanced the clarity and depth of our explanations accordingly. Furthermore, we will open-source our framework to support continued progress in this research area.*
>
> **Weaknesses & Questions**
>
> > **1&2. It is unclear what components are trainable if any are at all and how they are trained. This needs to be made explicit. Do the Proposed, Critic and Tool Planner share weights?**
>
> Our approach is built as a training-free agentic framework. This means we don't require any training data or fine-tuning throughout our pipeline. Instead, we leverage the robust capabilities of GPT-o3, a SOTA model from OpenAI known for its strong reasoning abilities.
>
> The Proposer, Critic, and Tool User agents in our framework are all instances of the same GPT-o3 model, which means they share the same underlying weights.
>
> However, each agent operates differently, as it is guided by a distinct input prompt tailored to define its specific role within our framework. Detailed prompt templates are provided in Appendix B of our paper.
>
> For example:
>
> - The Proposer is prompted to design a 3D tool asset based on a given robotic task description and 3D scene configuration. It outputs code for generating a 3D tool asset.
>
> - The rendered multi-view images of this tool, along with the Critic's system prompt, are then provided to the Critic agent. The Critic provides natural language modification suggestions, which feed back to the Proposer in an iterative refinement process until no more suggestions are needed.
>
> - The Tool User agent, prompted with the task, scene configuration, and the final tool description, outputs a robotic manipulation trajectory using a set of specified APIs (move, grasp, release). This is a one-time process.
>
> This strategy of repurposing a powerful, pre-trained model for various roles is a common and effective practice, as demonstrated in prior works like [1, 2, 3, 4].
>
> [1] Wang, Zihao, et al. "Describe, explain, plan and select: interactive planning with large language models enables open-world multi-task agents." Proceedings of the 37th International Conference on Neural Information Processing Systems. 2023.
>
> [2] Sarch, Gabriel, et al. "Vlm agents generate their own memories: Distilling experience into embodied programs of thought." Advances in Neural Information Processing Systems 37 (2024): 75942-75985.
>
> [3] Park, Joon Sung, et al. "Generative agents: Interactive simulacra of human behavior." Proceedings of the 36th annual acm symposium on user interface software and technology. 2023.
>
> [4] Huang, Wenlong, et al. "Inner Monologue: Embodied Reasoning through Planning with Language Models." Conference on Robot Learning. PMLR, 2023.
>
> > **3. The tool user is not mentioned in the problem setting/method overview.**
>
> Our pipeline consists of 3 components: a critic tool designer, a tool use planner, and a joint optimizer. Within the critic tool designer, there are two VLM agents, Proposer and Critic, that collaboratively design tools. Within the tool use planner, there is a single VLM agent called Tool User to function. Thanks for pointing out that we only mentioned Tool User in Section 3.3, not in the method overview. We'll make it clear in the revisions.
>
>
> > **4. What are the APIs mentioned in 3.3? how are these obtained? are they open source libraries? these need better explanation.**
>
> The APIs are indeed a minimal set of three: Grasp, Release, and Move, as listed in Section 3.3. We designed and implemented these specific robot control APIs, drawing inspiration from approaches found in prior work such as [5, 6]. Yes, we plan to open-source the implementation of all three APIs, with a particular focus on the robust Grasp API, to facilitate reproducibility and future research.
>
> [5] Wang, Yufei, et al. "RoboGen: Towards Unleashing Infinite Data for Automated Robot Learning via Generative Simulation." International Conference on Machine Learning. PMLR, 2024.
>
> [6] Liu, Peiqi, et al. "OK-Robot: What Really Matters in Integrating Open-Knowledge Models for Robotics." 2nd Workshop on Mobile Manipulation and Embodied Intelligence at ICRA 2024.
>
> > **5. The optimisation process is not explained at all in the joint optimiser. This needs to be significantly more elaborate. Preliminaries for the CMA-ES should be added to the main paper or appendix, as it is an important component of the pipeline.**
>
> Our joint optimizer employs Covariance Matrix Adaptation Evolution Strategy (CMA-ES), which is a powerful evolution strategy for numerical optimization of non-convex problems. At its core, CMA-ES operates by iteratively updating a multivariate normal distribution from which new candidate solutions are sampled. The mean and covariance matrix of this distribution are adapted based on the performance of the sampled solutions, allowing it to efficiently converge on optimal parameters even in complex landscapes.
>
> In our specific setting, the optimization process within the joint optimizer uses the task metric directly as the reward signal for CMA-ES (e.g., the distance between the cube and ideal position for the Reaching task; refer to Appendix C for more metrics). CMA-ES then optimizes two key aspects: the tool shape parameters and the robot control trajectories.
>
> We appreciate this valuable feedback. We will expand on the optimization process in Section 3.5 of the final paper, while also including necessary CMA-ES preliminaries in the appendix for improved clarity.
>
>
> > **6. Table 1 results appear impressive, but some further investigation is necessary. What dataset is used? is this publicly available and if not, will it be made publicly available?**
>
> We curated this benchmark, including 9 tasks encompassing a diverse range of materials and robotic skills. Yes, this benchmark, along with its detailed task configurations, will be made publicly available to the research community to support future work.
>
>
> > **7&8. The best and average of 8 runs is used, however these results are not very meaningful as they are (best could be an outlier). It would be much better to show mean +- std and since multiple runs are computed, statistical significance of results would be in order. This is the same for the ablation study.**
>
> We appreciate the feedback regarding the statistical reporting of our results and ablation study. Here we present the mean and standard deviation of the P score of the main result and ablation study as follows:
>
> | Method \ Task | Reach | Hold a Phone | Lift a Bowl | Lift a Piggy | Dough Calabash | Flatten Dough | Cut Dough | Fill a Bottle | Transport Water | Overall |
> | :------------ | :----: | :----: | :----: | :----: | :----: | :----: | :----: | :----: | :----: | :----: |
> | **No Tool**       | 0.00±0.00 | 0.02±0.03 | 0.13±0.13 | 0.00±0.00 | 0.33±0.13 | 0.12±0.07 | 0.48±0.34 | 0.04±0.02 | 0.00±0.00 | 0.12±0.08 |
> | **Retrieval**     | 0.03±0.04 | 0.52±0.30 | 0.14±0.12 | 0.36±0.37 | 0.15±0.09 | 0.31±0.16 | 0.65±0.27 | 0.05±0.04 | 0.26±0.24 | 0.27±0.18 |
> | **Meshy**         | 0.46±0.20 | 0.79±0.20 | 0.06±0.06 | 0.62±0.32 | 0.61±0.22 | 0.56±0.27 | 0.53±0.34 | 0.58±0.27 | 0.00±0.00 | 0.47±0.32 |
> | **Ours**          | 0.59±0.36 | 0.75±0.22 | 0.71±0.32 | 0.90±0.20 | 0.56±0.22 | 0.69±0.36 | 0.48±0.38 | 0.51±0.34 | 0.47±0.32 | 0.63±0.31 |
>
> | Method | Ours w/o Text-to-3D | Ours w/o Optimizer | Ours w/o Tool Opt | Ours|
> | :------------ | :----: | :----: | :----: | :----: |
> | **Score** | 0.46±0.24 | 0.10±0.07 | 0.32±0.25 | 0.63±0.31 |
>
> Our initial choice to report success rates and best scores was because most task performances are binary. For such discrete outcomes, standard deviation could be less informative. The best score specifically aimed to highlight our pipeline's full potential. As expected with binary scoring, the standard deviation will be relatively high for many tasks.
>
>
> *We hope that the provided new experiments and additional explanations have convinced you of the merits of our work. We wish they had addressed your concerns and turned your assessment to the positive side. If you have any more questions, please feel free to let us know during the rebuttal window.*
>
> *We appreciate your time! Thank you so much!*
>
> Best,
>
> Authors

---

> > ### Comment · Reviewer_Kk5s · 2025-08-04
> >
> > Thank you for the rebuttal and for providing clarifications to my questions and those of other reviewers.
> > Most of my concerns are addressed through the rebuttal; the only thing that seems not very convincing after the rebuttal is the method, i.e. proposing yet another tool-augmented MLLM, in a new domain, particularly when the API list is so limited. With that in mind, I will raise my rating as the results are convincing and the motivation for using an MLLM in robotics is very clear.

---

### Official Review · Reviewer_d7mA · 2025-07-01

**Clarity:** 4
**Significance:** 3
**Originality:** 3
**Rating:** 4
**Confidence:** 4

**Summary:**

The authors propose an automated pipeline that leverages vision-language models (VLMs) and physics simulation to design and use tools for robotic manipulation. This pipeline iteratively generates tool designs and corresponding robot actions through three main modules: (1) a Critic Tool Designer, where two VLM agents propose and evaluate tool designs; (2) a Tool Use Planner, which generates action plans using robot APIs; and (3) a Joint Optimizer, which refines both tool geometry and action trajectories.

Tools are constructed from a set of modular parts, represented using a constructive solid geometry (CSG)-like format. This representation improves the language model’s ability to interpret and reason about 3D geometries.

The authors conduct extensive experiments demonstrating that their method outperforms state-of-the-art approaches in tool design and use for robotic manipulation tasks.

**Questions:**

See weakness

**Ethical Concerns:**

["NO or VERY MINOR ethics concerns only"]

**Final Justification:**

Thank you for the rebuttal information provided. I will maintain my original score.

**Limitations:**

Yes

**Quality:**

3

**Strengths And Weaknesses:**

Strengths
- The authors propose a novel method for robotic tool design leveraging vision-language models (VLMs).
- They validate their approach through extensive experiments across multiple benchmarks.

Weaknesses:
- The Critic Tool Designer module consists of a Proposer and a Critic agent that iteratively generate and refine tool designs. However, the capabilities of these agents and their reasoning processes remain unclear. For example, the comment “the tool is long enough to avoid undesired contact” lacks supporting rationale. Please provide more detailed examples illustrating how VLMs or LLMs arrive at such suggestions. Additionally, presenting more diverse cases with corresponding reasoning chains behind specific tool modifications would better demonstrate the agents’ understanding and design capabilities.

- In the Tool Use Planner, the system infers grasp orientation and plans movements based on visual input, with the VLM analyzing images to guide actions. However, it is not clear how reliably the VLM can determine grasp orientations, especially across varying object geometries. More comprehensive quantitative and qualitative results are needed to evaluate the VLM’s performance on different object types and shapes, and to justify its effectiveness in planning grasp and motion strategies.

---

> ### Author Rebuttal · Authors · 2025-07-31
>
> *We appreciate the positive and insightful comments from you! We adress your concerns in details below.*
>
> **Weaknesses & Questions**
>
> **1. Detailed examples of Agents**
>
> To address the specific example, the comment about "the tool being long enough to avoid undesired contact" was made in the context of the "Transport Water" task. The rationale behind this design is to ensure the robot's gripper, which holds the tool, does not have to directly enter the water tank. In general, a handle of sufficient length is necessary to maintain a safe distance from the water and other materials, such as food, preventing contamination and protecting the robot's end-effector.
>
> We appreciate your suggestion to provide more detailed examples for a clearer understanding of our method's capabilities. We will add more detailed examples of chat history between Proposer and Critic to the appendix of the final paper, their reasoning chains.
>
> We appreciate your suggestion to provide more detailed examples and reasoning chains. We will add more demonstrations, including chat histories between the Proposer and Critic, and their corresponding reasoning chains, to the appendix of the final paper. For immediate illustration, here is a summary of a tool design case for the "Lift A Piggy" task, where the main challenge is that the piggy is large, round, and not directly graspable by the robotic gripper:
>
> (Due to rebuttal rules, images cannot be included, so we provide textual summaries of the shapes and suggestions.)
>
> Lift A Piggy Task: 4-Round Tool Design Modification Example
>
> 1. Proposer's Initial Design: A cube with a thin rod, looks like:
> ```
>      _________
>     /       /|
>    /_______/ |
>   |       |  |
>   |       | /
>   |_______|/
>        ||
>        ||
>        ||
>        ||
> ```
> 2. Critic's Suggestion (Round 1): Suggests redesigning the piggy bank interface. Specifically: "instead of a plain cube/rod, use an insert-and-twist key sized for the coin slot". (Note: The Critic may offer multiple suggestions, but the Proposer typically acts on the first one or two.)
> 3. Proposer's Modification (Round 2): Adds a bottom cube to the initial design, attempting to create a key-like structure:
> ```
>      _________
>     /       /|
>    /_______/ |
>   |       |  |
>   |       | /
>   |_______|/
>        ||
>        ||
>       _||__
>      / || /|
>     /___ / /
>     |____|/
> ```
> 4. Critic's Suggestion (Round 2 Continued): Suggests modifying the  newly added bottom cube: "Swap the bottom cube for a T-key/cam head that is narrow for insertion and wider when rotated ~90° to catch the inner rim."
> 5. Proposer's Modification (Round 3): Edits the shape parameters of the bottom cube to more closely resemble the T-key.
> ```
>      _________
>     /       /|
>    /_______/ |
>   |       |  |
>   |       | /
>   |_______|/
>        ||
>        ||
>        ||_
>       /||/|
>      /_ //
>      |_|/
> ```
> 6. Critic's Suggestion (Round 3 Continued): Suggests aking the top cube (the graspable part) thinner to facilitate holding.
> 7. Proposer's Modification (Round 4): Edits the shape parameters of the top cube to make it thinner.
> ```
>     _____________
>    /           /|
>   /___________/ |
>  |           | /
>  |___________|/
>        ||
>        ||
>        ||_
>       /||/|
>      /_ //
>      |_|/
> ```
> 8. Critic's Final Assessment: Done. No more modification suggestions.
>
> Additionally, we would like to direct you to the appendix B for the prompt templates of our agents, which offer further insight into how we guide their reasoning. We appreciate this feedback and will ensure the final version of the paper provides a more comprehensive view of our agents' design capabilities.
>
> The following is raw generated data from the Proposer and Critic to illustrate this process:
> ```
> Proposer (round 1)
> {
>     "name": "PiggyBank Lifter",
>     "parts": [
>         {
>             "geom": "cube",
>             "prompt": "",
>             "parameters": [
>                 0.04,
>                 0.04,
>                 0.04
>             ],
>             "is_graspable": true
>         },
>         {
>             "geom": "cylinder",
>             "prompt": "",
>             "parameters": [
>                 0.0025,
>                 0.05
>             ],
>             "is_graspable": false
>         }
>     ],
>     ......
> }
>
> Critic (Round 1)
> MODIFY
> Suggestions:
> 1. Redesign the piggy-bank interface at the opposite end: instead of a plain cube/rod, use an insert-and-twist key sized for the coin slot (tapered lead-in, soft pads, depth-stop collar, and a 90° cam or T-head that expands under the rim) so the spherical, smooth bank can be lifted without external grasping.
> 2. Add a dedicated gripper block at the top: a compact rectangular prism with two broad parallel faces (plus thin rubber pads or light knurling) so a parallel gripper can pinch it securely without slipping or rolling.
> 3. Strengthen the rod and joints: use a thicker shaft (≈8–12 mm), make the head and shaft one piece or mechanically keyed/pinned (avoid glue), and add fillets/gussets at both transitions to prevent bending or loosening under twist-and-lift loads.
> 4. Ensure alignment and balance: keep the shaft collinear with the insert head’s centroid so the load hangs vertically, place the gripper block centered above the shaft, and add simple orientation marks (“insert” vs. “locked”) to guide the twist.
> 5. Provide torque features & safety: give the gripper a small transverse bar or shallow finger groove on the block for better twist leverage, include a detent/stop to hold the locked position, and round all external edges to avoid scratching the bank.
>
> Proposer (round 2)
> {
>     "name": "PiggyBank Secure Lifter",
>     "parts": [
>         ......,
>         {
>             "geom": "cube",
>             "prompt": "",
>             "parameters": [
>                 0.02,
>                 0.02,
>                 0.005
>             ],
>             "is_graspable": false
>         }
>     ],
>     ......
> }
>
>
> Critic (Round 2)
> MODIFY
> 1. Insert-and-twist head: Swap the bottom cube for a T-key/cam head that is narrow for insertion and wider when rotated ~90° to catch the inner rim; include a tapered lead-in, soft sleeves, a depth-stop collar, and a positive rotation stop/detent so it stays locked during lift.
> 2. Shaft & joint reinforcement: Thicken the shaft (≈8–12 mm), use a keyed/D-flat connection to both the plate and the T-head (avoid plain round press-fits), and add fillets or a gusseted boss where the shaft meets the plate to prevent bending under twist-and-lift loads.
> 3. ......
>
> Proposer (Round 3)
> {
>     "name": "Secure PiggyBank Lifter",
>     "parts": [
>         ......,
>         {
>             "geom": "cube",
>             "prompt": "",
>             "parameters": [
>                 0.02,
>                 0.005,
>                 0.005
>             ],
>             "is_graspable": false
>         }
>     ],
>     ......
> }
>
> Critic (Round 3)
> MODIFY:
> 1. Gripper interface upgrade: Replace the small top block with a broad, flat plate that fits your gripper’s maximum aperture; keep two clean, parallel edges for pinching, add thin rubber pads for friction, chamfer the rim to avoid snagging, and use weight-relief cutouts so the plate is light yet stiff.
> 2. Improve the insert-and-twist head: Replace the tiny T-tab with a cam/T-key that is narrow for insertion and wider when rotated ~90° to catch the inner lip; include a depth-stop collar, soft sleeves for friction, rounded edges to avoid scratching, and a positive detent/stop so it stays locked under load.
> 3......
>
> The Proposer (Round 4)
> {
>     "name": "Optimized Piggy Bank Lifter",
>     "parts": [
>         {
>             "geom": "cube",
>             "prompt": "",
>             "parameters": [
>                 0.12,
>                 0.08,
>                 0.02
>             ],
>             "is_graspable": true
>         },
>         ......,
>     ],
>     ......
> }
>
> Critic (Round 4)
> DONE
> ```
>
> **2. Reliability of VLM Planner**
>
> Our tool-use planner is designed to efficiently solve simple tasks and provide strong initialization for complex ones. While grasp orientation is guided by a visual-language model (VLM) prior, the grasping pose itself is determined via sampling. We select the successful grasp with orientation closest to the VLM's suggestion. This approach ensures adaptability to varying geometries while leveraging the VLM's prior to accelerate sampling, without requiring perfect accuracy.
>
> In our experiments, removing the planner made it nearly impossible to discover successful trajectories within the time limit for most tasks.
>
> To further evaluate our grasping component, we tested the planner's ability to generate grasp orientations for 200 objects from a dataset. We measured the angular distance between the VLM-proposed orientation and the nearest feasible grasp found by our system. The average angular distance between our found grasp pose and the VLM's suggestion was 0.33. This compares favorably to an average distance of 0.87 when a working grasp pose was found via random sampling. These results confirm that the VLM provides useful priors, significantly improving our grasping pipeline's sampling efficiency.
>
> *We wish that our response has addressed your concerns. If you have any more questions, please feel free to let us know during the rebuttal window. Thank you very much! We appreciate your suggestions and comments! Thank you!*
>
> Best
>
> Authors

---

### Official Review · Reviewer_rFNK · 2025-07-03

**Clarity:** 3
**Significance:** 3
**Originality:** 3
**Rating:** 4
**Confidence:** 2

**Summary:**

The authors propose a pipeline for automatically generating suitable tools for robot manipulators in embodied agent tasks. The approach leverages LLM agents to collaboratively design initial tools using Constructive Solid Geometry (CSG) representations and generates initial action trajectories by minimizing trajectory length. To enable real-world applicability, the pipeline employs Evolution Strategies to jointly optimize both shape parameters and action trajectories. The effectiveness of the method is validated through both simulation and real-world experiments.

**Questions:**

In addition to the issues mentioned above, the reviewer have the following questions:
- Is it possible to generate concave-shaped tools? Figure 5 only shows convex examples.
- Why are only the move API parameters jointly optimized (line 203)? What about the grasp API parameters?
- Why is the "Dough Calabash" cut success rate for "retrieval" only 21%, which is even lower than using no tool?

**Ethical Concerns:**

["NO or VERY MINOR ethics concerns only"]

**Final Justification:**

Thanks to the additional information in the rebuttal stage, the reviewer raises the score of significance.

**Limitations:**

yes

**Quality:**

2

**Strengths And Weaknesses:**

Quality

The paper addresses a highly relevant and innovative problem in embodied agents, representing a significant step toward generalizable embodied intelligence. The effectiveness of the proposed method is demonstrated through several simulation and real-world experiments. However, the number of experiments is limited, and the tasks are relatively simple, which restricts the comprehensiveness of the evaluation. Although the method is claimed to benefit long-horizon tasks, the evaluation is still conducted on individual tasks. A major concern is the diversity of tool generation; only a few qualitative empirical visualizations are provided, making it difficult to assess the method’s full potential and limitations.

Clarity

The paper clearly presents the challenges, motivation, and design, making it easy to follow and highly readable. The contributions are well articulated. Figure 2 effectively illustrates the pipeline, and the experimental results are presented clearly, demonstrating the method’s effectiveness both qualitatively and quantitatively.

Significance

The inclusion of real-world experiments is a notable strength that enhances the significance of the work. However, some important issues remain unaddressed. The diversity of tool generation using the CSG representation is only partially demonstrated (as shown in Figure 5), and the performance boundaries of this representation are unclear. Additionally, CSG is a crucial step in tool generation and should be compared with other representations. The configuration of the robotic arm is also not discussed, suggesting that the method may be specific to a particular robot and may not generalize to other robotic arms.

Originality

The paper addresses an important and timely problem in embodied AI, which is also a key challenge for imitation learning in long-horizon tasks. The proposed solution is interesting, involving the interaction of two agents for tool design and the use of evolutionary algorithms to iteratively optimize both the tool design and action trajectories.

---

> ### Author Rebuttal · Authors · 2025-07-31
>
> *We thank the reviewer for the positive and constructive comments. We address your concerns in detail below.*
>
> **W1. "However, the number of experiments is limited, and the tasks are relatively simple, ..."**
>
> There's currently no large scale benchmark settings for tool generation and tool use tasks. Compared to previous work [1, 2, 3], our task settings are more diverse in quantity and semantic complexity, and we cover various deformable and fluid materials rather than only rigid objects.
>
> [1] Guo, Michelle, et al. "Learning to design 3d printable adaptations on everyday objects for robot manipulation." 2024 IEEE International Conference on Robotics and Automation (ICRA). IEEE, 2024.
>
> [2] Li, Mengxi, et al. "Learning Tool Morphology for Contact-Rich Manipulation Tasks with Differentiable Simulation." 2023 IEEE International Conference on Robotics and Automation (ICRA). IEEE, 2023.
>
> [3] Liu, Ziang, et al. "Learning to Design and Use Tools for Robotic Manipulation." 7th Annual Conference on Robot Learning.
>
> **W2. "... the evaluation is still conducted on individual tasks."**
>
> Our approach aligns with common practice in robotics, where complex, long-horizon problems are often decomposed into manageable sub-tasks, as demonstrated by prior work like [4, 5].
>
> Our primary focus lies in validating our method on these fundamental individual tasks, rather than task decomposition or high-level planning. To further illustrate our method's capacity for complex, integrated scenarios, we've included a complete long-horizon task demonstration in Figure 1.
>
> [4] Wang, Yufei, et al. "RoboGen: Towards Unleashing Infinite Data for Automated Robot Learning via Generative Simulation." International Conference on Machine Learning. PMLR, 2024.
>
> [5] Huang, Zhiao, et al. "Diffvl: Scaling up soft body manipulation using vision-language driven differentiable physics." Advances in Neural Information Processing Systems 36 (2023): 29875-29900.
>
>
> **W3. "A major concern is the diversity of tool generation, ..." "The diversity of tool generation using the CSG representation is only partially demonstrated ..."**
>
> As shown in Figure 5, we've demonstrated the diversity of generated tools for two distinct tasks. To further address this concern, we can include additional examples of diverse tools generated for other tasks in the appendix of the final version.
>
> Compared to prior work like [3], we've significantly extended tool diversity in two key ways: by introducing a text-to-3D generation module for increased component complexity, and by designing a set of tool design APIs for enhanced connection diversity that allow 3D parts to be assembled into a tool.
>
> As for Constructive Solid Geometry (CSG), our tool representation is indeed inspired by CSG principles. However, we employ an implicit approach using 3D editing APIs for construction rather than explicit CSG operations. While CSG is a powerful method and widely used in CAD, its primary limitation is that it relies on a pre-defined set of primitives. We address this by complementing our approach with off-the-shelf 3D generation, bringing in complex, semantically meaningful meshes to fill in where CSG might fall short. We believe this mix gives us a solid balance between structured representation and generative flexibility.
>
> [3] Liu, Ziang, et al. "Learning to Design and Use Tools for Robotic Manipulation." 7th Annual Conference on Robot Learning.
>
> **W4. "The configuration of the robotic arm is also not discussed, ..."**
>
> The tool design is task specific, and the robot arm configuration is part of the task settings. For instance, a "reaching" task becomes trivial with a longer arm, just as a "lifting bowl" task might not require a tool if a dexterous hand is available. Currently, we mostly focus on robot arms with parallel grippers and leave the dexterous hands for future works.
>
> It's important to note that our designed tool adapts to the robot arm's configuration during the joint optimization process. For example, if you were to use a shorter arm for the reaching task, our method would optimize for a longer tool to compensate, effectively adapting to the specific arm in use.
>
> **Questions**
>
>
> > **1. Is it possible to generate concave-shaped tools? Figure 5 only shows convex examples.**
>
> Yes, our method is capable of generating concave tools. We can achieve this either by directly generating such shapes through the text-to-3D model (Meshy), as seen with the tool for "Hold A Phone" in Figure 3, or by explicitly creating them using a set minus operation within our tool design APIs. For instance, the "Dough Calabash" tool in Figure 3 was formed by subtracting a calabash shape from a cube, clearly demonstrating a concave design.
>
>
> > **2. Why are only the move API parameters jointly optimized (line 203)? What about the grasp API parameters?**
>
> The *Release* API has no parameters to optimize. For the *Grasp* API, we exclude its parameters from joint optimization to reduce computational burden, as stable grasping can be immediately evaluated.
>
> For the *euler* parameter in *Grasp* API, we specifically leverage the VLM's prior knowledge to guide grasping orientation, preventing awkward poses that could cause arm twisting, or self-collision during function.
>
> Given that there's often little practical difference between similar grasp orientations (e.g., 90° vs. 89° or even 90° vs. 70°), our *Grasp* implementation samples and evaluates numerous poses within a range of the VLM's ideal. So the VLM's exact orientation is not strictly guaranteed. Optimizing this *euler* parameter could confuse the optimizer, leading to less stable results.
>
> > **3. Why is the "Dough Calabash" cut success rate for "retrieval" only 21%, which is even lower than using no tool?**
>
> To clarify, the reported success rates for both "retrieval" and "no tool" are actually 0.0%. The values 0.56 and 0.21 represent the best score during 8 trials. The score here is the normalized CLIP scores between rendered image of the dough and the textual prompt "calabash". A score over 0.8 is considered a success for this task, and neither method reached that threshold.
>
> For the "no tool" scenario, the robot's gripper itself acts as a tool, allowing for various manipulations like pressing and pinching. In the "retrieval" case, the system attempted to retrieve either a hammer or a knife. These tools proved even less flexible than the gripper for shaping the dough, leading to lower performance. Ultimately, creating a true calabash shape remains a significant challenge for both approaches.
>
> *We wish that our response has addressed your concerns, and convinced you of the merits of our work. If you have any more questions, please feel free to let us know during the rebuttal window. Thank you very much!*
>
> *We appreciate your suggestions and comments! Thank you!*
>
> Best,
>
> Authors

---

### Note · Authors · 2025-08-12

We thank all the reviewers for their time and effort. Reviewer BMgf and Reviewer Kk5s found our rebuttal to have resolved their concerns and consider to raise their scores. Although we haven't yet received a response from Reviewer rFNK and Reviewer d7mA, their initial reviews were positive, and we hope our responses have fully addressed their concerns.

We are glad that the reviewers generally recognized our key contributions:

* An **innovative framework** for **general-purpose robotic tool design**.
* A **collaborative agent system** that effectively addresses this problem.
* **Extensive experiments** demonstrating our method's effectiveness.


To address the reviewers’ concerns, we've added new experiments:

* **\[A] Grasping-prior effectiveness:** We quantify how our grasping function—which leverages a VLM to approximate a grasping direction—improves the grasping process. \[Reviewer d7mA]
* **\[B] Significance measurement:** We report the **mean** and **standard deviation** of the **standard score** in our experiments. \[Reviewer Kk5s]
* **\[C] Failure analysis:** We identify failure cases, analyze their underlying causes, and report their proportions. \[Reviewer BMgf]
* **\[D] Real-world success rate:** We measure success rates in real-world experiments and identify the causes of failure. \[Reviewer BMgf]

We clarified several implementation details as requested by reviewers:

* **\[A]** **Real-world experiments** were conducted by transferring successful simulated designs and trajectories to physical robots. This validates the structural stability and physical feasibility of our tools. \[Reviewer BMgf]
* **\[B]** We included sample logs of the **proposer** and **critic** agents, showing the critic's **reasoning chain** to highlight their design capabilities. \[Reviewer d7mA]
* **\[C]** The **tool-use planner** uses a VLM to find a **prior grasping orientation** but still relies on **grasp-pose sampling** to find the final pose. This ensures **semantically correct** grasps that do not hinder tool use. \[Reviewers d7mA, BMgf, Kk5s]
* **\[D]** Our tool generation combines a **Constructive Solid Geometry (CSG)**-inspired approach with **3D editing APIs** and **text-to-3D generation** for richer geometry. \[Reviewers rFNK, BMgf]

Best,

Authors

---

### Decision · Program_Chairs · 2025-09-17

**Decision:**

Accept (poster)

**Comment:**

This paper develops an agentic approach to design and use tools for robotic manipulation, combining VLM-based tool design with evolutionary optimization. Reviewers were generally positive on the paper's clarity, the breadth of evaluation tasks (including deformables and fluids), the approach combining generative priors with physics-based optimization, and overall performance of the approach. The rebuttal clarified some aspects, added real-world success rates, and added ablations and additional analysis. In particular, this converted most of the initial concerns mentioned in the reviews to more minor limitations of the approach (e.g., minor novelty concerns, evaluation scale, and the extent of the real-world experiments). The authors are encouraged to incorporate the rebuttal and discussion when revising their paper.